# Distributionally Robust Graph Out-of-Distribution Recommendation via Diffusion Model

## Abstract

The distributionally robust optimization (DRO)-based graph neural network methods improve recommendation systems' out-of-distribution (OOD) generalization by optimizing the model's worst-case performance. However, these studies fail to consider the impact of noisy samples in the training data, which results in diminished generalization capabilities and lower accuracy. Through experimental and theoretical analysis, this paper reveals that current DRO-based graph recommendation methods assign greater weight to noise distribution, leading to model parameter learning being dominated by it. When the model overly focuses on fitting noise samples in the training data, it may learn irrelevant or meaningless features that cannot be generalized to OOD data. To address this challenge, we design a **D**istributionally **R**obust **G**raph model for **O**OD recommendation (DRGO). Specifically, our method first employs a simple and effective diffusion paradigm to alleviate the noisy effect in the latent space. Additionally, an entropy regularization term is introduced in the DRO objective function to avoid extreme sample weights in the worst-case distribution. Finally, we provide a theoretical proof of the generalization error bound of DRGO as well as a theoretical analysis of how our approach mitigates noisy sample effects, which helps to better understand the proposed framework from a theoretical perspective. We conduct extensive experiments on four datasets to evaluate the effectiveness of our framework against three typical distribution shifts, and the results demonstrate its superiority in both independently and identically distributed distributions (IID) and OOD. Our code is available at https://anonymous.4open.science/r/DRGO-FED2.

## CCS Concepts

• **Information systems → Recommender systems**.

## Keywords

Graph Recommendation, Distributionally Robust Optimization, Out-of-Distribution

**ACM Reference Format:**

Anonymous Author(s). 2018. Distributionally Robust Graph Out-of-Distribution Recommendation via Diffusion Model. In *Proceedings of Make sure to enter the correct conference title from your rights confirmation emai (Conference acronym 'XX)*. ACM, New York, NY, USA, 13 pages. https://doi.org/XXXXXXX.XXXXXXX

## 1 Introduction

Recently, Graph Neural Network (GNN)-based recommendation methods have gained extensive attention in various recommendation tasks and scenarios [3, 9, 12, 41, 46]. Generally, GNN-based methods learn embeddings of users or items by capturing high-order collaborative signals from the user-item interaction graph. Although these GNN-based methods have achieved state-of-the-art performance in the collaborative recommendation, some assume that the training and test sets follow an independent and identical distribution (IID). Unfortunately, this assumption may not hold when faced with out-of-distribution (OOD) data (i.e., distribution shift) in real-world recommendation scenarios. Such data distribution shifts may arise from various factors, such as changes in user consumption habits influenced by seasons, holidays, policies, etc. Additionally, various biases (e.g., popularity bias and exposure bias) in the recommendation system can lead to inconsistencies between training and test data distributions.

Existing works have attempted to address the challenge of distribution shift using various techniques to enhance the generalization of recommendation models. For instance, some works [5, 24, 42] use causal learning and approximate inference methods to infer environment labels and further learn representations that are insensitive to the environment. Some works [30, 38] achieve OOD generalization by directly decoupling user-variant and invariant representations. In addition, other works employ data augmentation or self-supervised learning [31, 37] methods to enhance the generalization performance of GNN-based methods on OOD data. However, the methods mentioned above also have some limitations in addressing distribution shifts: (1) The performance of these methods, such as [5, 24, 42], relies on explicit environments as labels. If the environmental factors cannot be inferred, achieving satisfactory generalization will be difficult. And these methods [30, 38] to disentangle variations from invariant preferences lack theoretical guarantees. (2) These methods [31, 37, 37] target a specific type of bias, such as popularity bias, and cannot address scenarios with multiple complex data distribution shifts.

Recently, to address the limitations of these methods, researchers are leveraging Distributionally Robust Optimization (DRO) [20] to enhance recommendation models' ability to generalize beyond their training data [23, 35, 43]. DRO focuses on improving model robustness by considering a set of potential distributions and optimizing performance for the worst-case scenario among them. This approach helps ensure the model performs well even when facing distribution shifts or operating in new environments. Despite the success, our experimental and theoretical analyses suggest that existing DRO-based recommendation methods still face the following two challenges:

- **Challenge 1**: Existing DRO-based recommendation methods [23, 43] assign greater weight to noise distribution when searching for the worst-case scenario. This causes model parameter learning

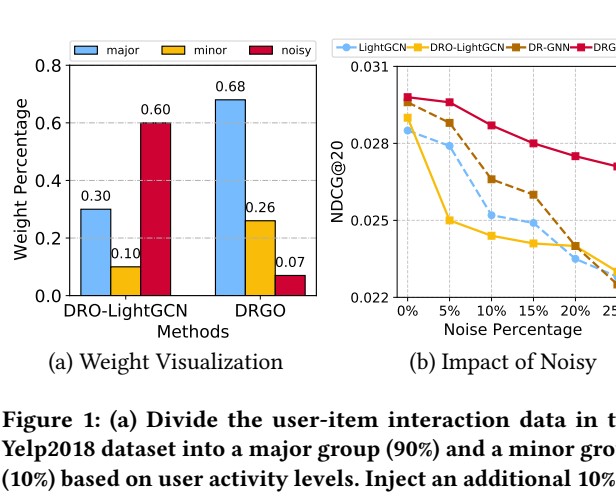

(a) Weight Visualization

(b) Impact of Noisy

**Figure 1: (a) Divide the user-item interaction data in the Yelp2018 dataset into a major group (90%) and a minor group (10%) based on user activity levels. Inject an additional 10% of the data as a noise group. Compare the performance of DRO-LightGCN and our DRGO, observing the weight changes for each group across multiple iterations. (b) The model's performance under different noise levels on the Yelp2018 dataset.**

to be dominated by noise, leading to overfitting on irrelevant features and further reducing generalization on OOD data.

- **Challenge 2**: Certain DRO methods based on the Kullback-Leibler (KL) divergence [23, 35] assume that there is an overlap between the distributions of prior interactions and the testing phase. However, due to the randomness and unpredictability of user behavior, the overlap between the test distribution and the training distribution may be minimal or even nonexistent, which makes this assumption limited. This situation hinders the effective computation of the KL divergence, thereby restricting the robustness and performance of the model.

For the first challenge, as illustrated in Figure 1, we conduct experiments to verify how noise samples affect the generalization performance of the DRO model. Figure 1(a) illustrates the weight changes across different groups after several training iterations, revealing that the performance of DRO-LightGCN [23, 43] is affected by the noise distribution. Figure 1(b) shows performance variations under different noise ratios, indicating that when the noisy ratio exceeds 20%, the performance of DRO-LightGCN falls behind that of LightGCN. Although DR-GNN is also affected by noisy data, its data augmentation component can alleviate the impact of noise to some extent. Section 3.1 provides a more detailed theoretical analysis of the challenges mentioned above.

Considering the limitations and challenges of current solutions, we introduce a novel Distributionally Robust Graph framework, termed DRGO, aimed at improving the robustness and generalization of recommendation systems when dealing with OOD data. To address the first challenge, our DRGO utilizes a graph variational autoencoder to encode the feature structure of the graph into a fixed distribution. Then, a diffusion model is employed in a low-dimensional embedding space to alleviate the impact of noise samples in the interaction data. We also introduce an entropy regularization term into the objective function of DRGO to prevent the model from optimizing over extreme distributions. For the second challenge, we attempt to utilize Sinkhorn DRO as a replacement for KL-based DRO. Sinkhorn DRO maintains model robustness in

non-overlapping or shifted distributions, ensuring minimal performance degradation in worst-case scenarios. We rigorously provide the theoretical proof of the generalization bound of our DRGO on OOD data, as well as the theoretical analysis of how our DRGO mitigates the impact of noisy samples.

The main contributions of our paper are summarized as follows:

- **Motivation**. Through experimental and theoretical analysis, we reveal the limitations and shortcomings of DRO-based graph recommendation methods in addressing data distribution shifts.
- **Methodology**. This work proposes a novel Distributionally Robust Graph-based method for OOD recommendation called DRGO. It integrates diffusion models and an entropy regularization term to remove noise while reducing focus on extreme distributions (such as noise distribution). Additionally, we provide rigorous theoretical guarantees for DRGO.
- **Experiment**. We conduct extensive experiments to validate the superiority of our proposed method under three types of distribution shifts: popularity, temporal, and exposure. The results confirm the model's exceptional performance on both OOD and IID data.

## 2 Preliminary

In this section, we first define the notations used in this paper and then introduce the definition of DRO. Due to page limitations, we have provided an introduction to Diffusion model in Appendix E.

### 2.1 Problem Definition

**Recommendation on Graphs**. In GNN-based recommendation methods, we consider a given user-item interaction matrix $\mathbf{R} \in \mathbb{R}^{|\mathcal{U}| \times |\mathcal{V}|}$, where $\mathcal{U} = \{u_1, u_2, \ldots, u_m\}$ denotes the user set and $I = \{i_1, i_2, \ldots, i_n\}$ represents the item set. GNN-based methods first convert the interaction matrix into graph-structured data $\mathcal{G} = \{\mathcal{U}, I, \mathcal{E}\}$ and the $\mathcal{E} = (u, i)|r_{u,i} = 1$ is the edge set. GNN-based methods use $\mathcal{G}$ as input to capture higher-order collaborative signals between users and items, thereby inferring user preferences towards items.

**Distribution shift on Recommendation**. Due to environmental factors, training and testing data distributions may differ in real-world scenarios. The environment is a key concept, as it is the direct cause of data-generating distributions. For example, in recommendation scenarios, weather, policy changes, and user mood can be considered environments. These factors can lead to changes in user behavior, resulting in distribution shifts in the data. Most traditional recommendation models trained under the IID assumption fail to generalize OOD data well.

### 2.2 Distributionally Robust Optimization

Distributionally Robust Optimization (DRO) differs from other OOD recommendation methods [5, 24, 42] that require environmental labels. Its core idea is to identify and optimize for the worst-case data distribution within a predefined range of distributions. This approach ensures that the model maintains good performance when facing distribution changes or shifts, thereby improving the model's generalization performance on OOD data. DRO is typically solved through a bi-level optimization problem: the outer optimization determines the model parameters that perform best under the worst

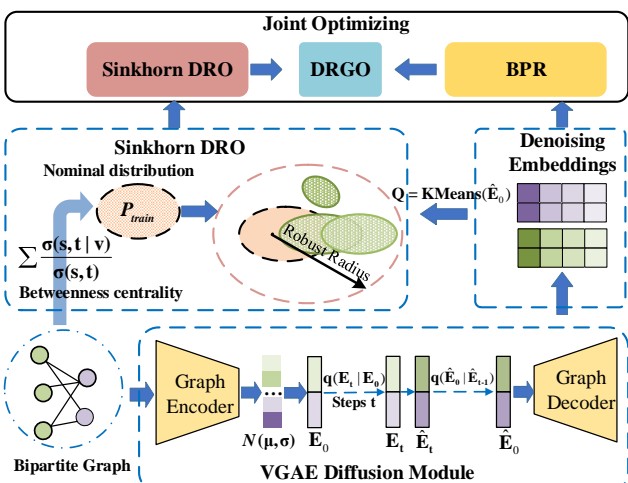

**Figure 2: The proposed DRGO model schematic. An bipartite graph is first processed using VGAE diffusion to obtain denoised embeddings. Then, betweenness centrality and $k$-means clustering are applied to construct the nominal distribution and uncertainty set. Finally, a joint optimization strategy is employed for optimization.**

distribution, while the inner optimization adjusts for all possible distributions. Its mathematical formulation is as follows:

$$\min_{\theta \in \Theta} \left\{ R(\theta) := \sup_{Q \in \mathcal{P}(P_{train})} \mathbb{E}_{(x,y) \sim Q} \left[ \mathcal{L}(f_\theta(x), y) \right] \right\} \tag{1}$$
$$\text{s.t. } D(Q, P_{train}) \le \rho,$$

where $\theta$ represents the optimal model parameters in DRO, and $\mathcal{L}(\cdot)$ denotes the loss function. sup is the supremum, which means finding the maximum value under the given conditions. The uncertainty set $Q$ approximates potential test distributions. It is typically defined as a divergence ball with radius $\rho$ centered around the nominal distribution $P_{train}$, expressed as $Q = \{Q : D(Q, P_{train}) \le \rho\}$, where $D(\cdot, \cdot)$ represents a distance metric such as Wasserstein distance or Kullback-Leibler (KL) divergence. Generally, constructing the uncertainty set and the nominal distribution is crucial to DRO.

## 3 Methodology

In this section, we first discuss the limitations of existing DRO-based methods, then propose a novel DRO approach and present the complete model structure, followed by a theoretical analysis. Figure 2 and Algorithm 1 (in Appendix) show the overall framework of the model and the algorithm pseudocode.

### 3.1 Limitations of DRO-based Recommendation Method

**Drawback of KL divergence**. Some recent works [23, 35] assume an overlap between the distributions of the training and testing sets and use KL divergence as a constraint for the uncertainty set. However, in practical scenarios, user behavior may exhibit high levels of uncertainty and significant randomness, leading to situations

where the distribution of the testing set shares no standard support with that of the training set. KL divergence becomes extremely large in such cases, rendering the model optimization meaningless and preventing effective OOD generalization. We give a detailed theoretical derivation in the Appendix A.1.

**Impact of noise samples**. Without loss of generality, we use the Bayesian Personalized Ranking (BPR) [21] loss, a commonly used ranking function, to analyze the impact of noise samples on recommendation performance. Specifically, DRO can be viewed as optimizing over a weighted empirical distribution. Following the method proposed in [14], we demonstrate from the perspective of model parameter variance how noise influences the learning of DRO model parameters through weighted BPR loss. In the BPR loss function, the variance of the model parameters reflects the model's sensitivity to various samples, especially when the training data contains noise. This sensitivity can lead to increased variance and instability of the model. Due to page limitations, please refer to Appendix A.2 for more detailed theoretical derivation. Based on the weighted BPR loss function and the derived variance of the model parameters, we can get the following conclusions:

- If the weight of noisy samples $w_o(j)$ is greater than the weight of clean samples $w_c(i)$, then noisy samples will have a more significant impact on the estimate of $\theta$. This can lead to an increase in the variance of parameter estimates, causing the model to become unstable. This instability can potentially reduce the model's generalization ability on OOD data.
- Even if the proportion of noisy samples is small, the more significant variance $\sigma_o^2$ of noisy samples can still significantly affect the parameter estimates, leading to overfitting to the noisy samples.

Overall, we can understand that in DRO-based methods, the impact of noise distribution cannot be ignored when improving the OOD generalization performance of recommendations. Otherwise, the model's performance will be limited.

### 3.2 Sinkhorn DRO

To address the challenge of limited OOD performance in KL divergence based DRO methods, this paper introduces Sinkhorn Distributionally Robust Optimization (i.e., Sinkhorn DRO). Sinkhorn DRO combines distributionally robust optimization with the Sinkhorn distance to tackle robustness optimization problems in distributional shifts. Sinkhorn distance is a variant of the Wasserstein distance, smoothed by entropy regularization. It is defined as follows:

$$W_{c,\lambda}(P, Q) = \inf_{\pi \in \Pi(P,Q)} \mathbb{E}_{(x,y) \sim \pi} \left[ c(x, y) \right] + \lambda \cdot H(\pi), \tag{2}$$

where $\Pi(P, Q)$ denotes the set of all joint distributions whose marginal distributions are $P$ and $Q$, i.e., all possible transportation plans from distribution $P$ to distribution $Q$. inf represents infimum. $\pi$ is the entropy of the joint distribution $\pi$, and $\lambda > 0$ is the regularization parameter. We further provide the mathematical definition of Sinkhorn DRO as follows:

$$\min_{\theta \in \Theta} \left\{ R(\theta) := \sup_{Q: W_{c,\lambda}(P_{train}, Q) \le \rho} \mathbb{E}_{P_{train}(x,y) \sim Q} \left[ \mathcal{L}(f_\theta(x), y) \right] \right\} \tag{3}$$

where $W_{c,\lambda}(P_{train}, Q)$ is the Sinkhorn distance between the nominal distribution $P_{train}$ and the test distribution $Q$, and $\rho$ is a radius that controls the range of robust optimization. By introducing entropy regularization, Sinkhorn distance avoids the issue of KL divergence becoming infinite when distributions do not overlap, allowing the model to perform reasonable optimization through geometric relationships even when no data in the training set is similar to the test set.

## 3.3 Model Instantiations

In this section, we detail how our proposed DRGO method leverages diffusion models and DRO to mitigate the limited model generalization caused by noisy samples in training data.

*3.3.1 Denoising Diffusion Module.* We are motivated by the effectiveness of diffusion models in producing clean data across diverse areas, including images [47] and text [17]. Our proposed DRGO method incorporates diffusion models to generate denoised user-item embeddings. Considering the discrete nature of user-item interaction graphs, they are unsuitable for direct input into diffusion models. Therefore, we design a method that integrates Variational Graph AutoEncoders (**VGAE**) [8] with diffusion models to efficiently denoise graph-structure data. In detail, consider a user-item interaction graph $\mathcal{G}_o = \{\mathcal{U}, \mathcal{V}, \mathcal{E}, \mathcal{X}\}$, and $\mathcal{X}$ denotes the features of the nodes, such as a user's gender and age or an item's category. Existing work [13] suggests that leveraging stable features related to the target distribution can effectively improve a model's generalization performance on OOD data. We use $\mathcal{G}_o$ as the input to the VGAE model, mapping it to a low-dimensional latent vector $\mathbf{E}_0 \sim \mathcal{N}(\mu_0, \sigma_0)$ to facilitate both forward and reverse diffusion processes in the latent space. The encoding process is given by:

$$q_\psi(\mathbf{E}_0 | \mathcal{A}, \mathcal{X}) = \mathcal{N}(\mathbf{E}_0 | \mu_0, \sigma_0), \qquad (4)$$

where $\mathcal{A}$ represents the adjacency matrix of $\mathcal{G}_o$, $\mu_0 = GCN_\mu(\mathcal{A}, \mathcal{X})$ is the matrix of mean vectors, and $\sigma_0 = GCN_\sigma(\mathcal{A}, \mathcal{X})$ denotes the matrix of standard deviations. Here, $GCN(\cdot)$ refers to the graph convolutional network in the graph variational autoencoder. Using the reparameterization trick, $\mathbf{E}_0$ is calculated as:

$$\mathbf{E}_0 = \mu_0 + \sigma_0 \odot \epsilon, \qquad (5)$$

where $\epsilon \sim \mathcal{N}(0, I)$ and $\odot$ denotes element-wise multiplication. $\mathbf{E}_0$ will be used as the input to the diffusion model for denoising, and the entire process can be represented as:

$$\mathbf{E}_0 \xrightarrow{\phi} \cdots \mathbf{E}_t \xrightarrow{\psi} \hat{\mathbf{E}}_{-1} \cdots \xrightarrow{\psi} \hat{\mathbf{E}}_0, \qquad (6)$$

where $\phi$ represents injecting noise into the input embedding, while $\psi$ removes the noise through optimization. After obtaining the denoised embedding, the decoder of the VGAE takes $\hat{\mathbf{E}}_0$ as input to reconstruct $G_0$, which can be represented as: The entire process is illustrated as follows:

$$\hat{\mathcal{A}} = \sigma(\hat{\mathbf{E}}_0 \hat{\mathbf{E}}_0^\top), \qquad (7)$$

where $\sigma(\cdot)$ is the sigmoid function. The variational lower bound optimizes the object of VGAE is shown as:

$$\begin{aligned} \mathcal{L}_{VGAE} = \; & \mathbb{E}_{q(\mathbf{E}_0|\mathcal{A},\mathcal{X})} \left[ \log p_\theta(\hat{\mathcal{A}}|\mathcal{X}) \right] \\ & - D_{KL}\left( q(\mathbf{E}_0|\mathcal{A},\mathcal{X}) \parallel p(\mathbf{E}_0) \right), \end{aligned} \qquad (8)$$

where $q$ represents the approximate posterior distribution, while $p$ represents the reconstruction probability distribution. Additionally, the optimization objective of the diffusion model is shown in Eq. (57). We will elaborate on the optimization objective of DRGO in detail in Section 3.3.4.

*3.3.2 Nominal Distribution.* Typically, when using DRO to optimize on OOD data, the nominal distribution $P_{train}$ should ideally cover the testing distribution within a radius $\rho$. However, in practical recommendation scenarios, due to the unpredictability and randomness of user behavior, the distribution of the test set is often unknown, posing challenges for constructing the nominal distribution. In reality, popular items or users in recommendation scenarios often influence the future behavior of other users. Considering the connectivity of graph-structured data, this study leverages betweenness centrality to construct the nominal distribution. **The basic assumption is that groups with high centrality during training strongly influence unseen distribution groups** [2, 18]. Betweenness centrality measures the frequency at which an entity appears on the shortest paths between other entities in a topology. Entities with higher betweenness centrality have greater control over the topology. Its mathematical definition is as follows:

$$c_v^{data} = \sum_{s,t \in E_{\text{train}}} \frac{\sigma(s, t \mid v)}{\sigma(s, t)}, \qquad (9)$$

where $\sigma(s, t)$ is the number of shortest paths between groups $s$ and $t$ in the graph (i.e., $(s, t)$-paths), and $\sigma(s, t \mid v)$ is the number of $(s, t)$-paths that pass through node $v$ in graph $\mathcal{G}_0$. After obtaining the betweenness centrality values for each node, we sort them in descending order and use the embeddings of the top-$n\%$ nodes to construct our nominal distribution. In the hyperparameter analysis experiments, we will discuss in detail the impact of the value of $n$ on model performance.

*3.3.3 Uncertainty Set.* When using DRO-based methods, it is necessary to assign weights to different groups to optimize the worst-case scenario. However, an excessive number of groups can make DRO optimization challenging. Drawing on previous work [45], we can directly cluster the denoised user embeddings $\hat{E}_u$, thereby reducing the number of samples. Additionally, clustering algorithms can more accurately group users with similar behaviors by leveraging user feature information. Formally, the uncertainty set $Q$ can be obtained as follows:

$$Q = \{q_1, q_2, \ldots, q_k\} = \text{KMeans}(\hat{E}_u), \qquad (10)$$

where the number of clusters $k$ will be discussed in detail as a hyperparameter.

*3.3.4 Joint Optimization.* We propose the DRGO method, which uses LightGCN as the backbone and employs Bayesian Personalized Ranking as the optimization objective for the recommendation task:

$$\mathcal{L}_{rec} = \sum_{(u,i^+,i^-)} -\log \sigma(\hat{r}_{u,i^+} - \hat{r}_{u,i^-}), \qquad (11)$$

where $(u, i^+, i^-)$ is a triplet sample for pairwise recommendation training. $i^+$ represents positive samples from which the user has interacted, and $i^-$ are the negative samples randomly drawn from the items with which the user has not interacted, respectively. $\hat{r}_{u,i^+}$ represents the positive prediction score and $\hat{r}_{u,i^-}$ is the negative

prediction score. Therefore, the overall optimization objective of our DRGO is as follows:

$$\min_{\theta} \left\{ R(\theta, w_{q_i}) := \sup_{Q:W_{c,\lambda}(P_{train}, Q) \leq \rho} \sum_{i=1}^{k} w_{q_i} \mathcal{L}_{rec}(f(x), y) \right.$$

$$\left. - \beta \sum_{i=1}^{k} w_{q_i} \log w_{q_i} + \mathcal{L}_{denoising}(\mathcal{G}_0) \right\}, \tag{12}$$

where $w_{q_i}$ represents the weights of different groups $q_i$, which need to be dynamically adjusted during training. The parameter $\beta$ serves as the penalty factor coefficient. The term $\sum_{i=1}^{n} w_{q_i} \log w_{q_i}$ represents an entropy regularization component that encourages the distribution $q$ to approach uniformity and $k$ is the number of groups (i.e, the number of clusters $k$). This promotes balanced weighting of all samples during optimization and helps prevent the model from disproportionately focusing on outliers, such as noisy samples. The term $\mathcal{L}_{denoising}$ represents the loss function of the denoising module, primarily comprising the loss from the VGAE, as defined in Eq. (8), and the diffusion model, as defined in Eq. (57), which is displayed as:

$$\mathcal{L}_{denoising} = \mathcal{L}_{VGAE} + \mathcal{L}_{sample} \tag{13}$$

## 3.4 Theoretical Analysis

In this subsection, we primarily provide two theoretical analyses: (1) the generalization bound of DRGO on OOD data and (2) how DRGO mitigates the impact of random noise.

THEOREM 3.1. *Assume the loss function $R(\theta, W_{q_i})$ is bounded by a constant B. For $\delta > 0$, with probability at least $1 - \delta$, the following inequality holds for all $f \in F$:*

$$\hat{R}(\theta, w_{q_i}) \leq R(\theta, w_{q_i}) + 2\hat{R}_n(F) + BW(P_{train}, Q) + B\sqrt{\frac{ln(1/\sigma)}{2n}}. \tag{14}$$

The upper bound of the generalization risk on $Q$ is primarily influenced by its distance to $P_{train}$, denoted as $W(P_{train}, Q)$. By incorporating the topological prior, the risk on $Q$ can be tightly constrained by minimizing $W(P_{train}, Q)$, provided that $Q$ lies within the convex hull of the training groups. We experimentally validate the effectiveness of the topological prior in reducing generalization risks across unseen distributions. We provide a complete theoretical proof in Appendix A.3.

We analyze and discuss from the perspective of DRGO's gradient updates how DRGO mitigates the impact of random noise, focusing primarily on the gradients of the second and third terms in the DRGO optimization objective.

The introduction of the negative entropy term is aimed at regularizing the weights $w_{q_i}$, preventing certain samples (especially noisy) from receiving excessively high weights during optimization. The gradient of the negative entropy term is:

$$\nabla_{w_{q_i}} \left( -\sum_{i=1}^{k} w_{q_i} \log w_{q_i} \right) = (-\log w_{q_i} + 1) \tag{15}$$

This gradient has the following properties: When the weight $w_{q_i}$ is large, the gradient of the negative entropy term is also large, thereby reducing the weight. When the weight $w_{q_i}$ is small, the

influence of the negative entropy term on the weight is minimal, ensuring a balanced distribution of sample weights.

The denoising loss $\mathcal{L}_{denoising}$ introduces a diffusion mechanism to gradually restore noisy samples to a noise-free state. The gradient of this loss is:

$$\nabla_{\theta} \mathcal{L}_{denoising} = \mathbb{E}_{\tilde{x}} \left[ 2 \left( G_0(\tilde{x}) - x \right) \cdot \nabla_{\theta} G_0(\tilde{x}) \right] \tag{16}$$

where $\tilde{x}$ represents the noisy input sample, and $G_0(\tilde{x})$ is the denoised output of the model. The objective is to make the denoised sample $G_0(\tilde{x})$ as close as possible to the original sample $x$. Noise Mitigation Mechanism: The denoising loss calculates the difference between the denoised sample and the original sample, adjusting the gradient of noisy samples in the direction of denoising, thereby reducing the interference of noise in the gradient update. During the denoising process, the impact of noisy samples is gradually reduced, causing the model's gradient updates to rely more on noise-free samples, thus improving the model's robustness in noisy environments.

In summary, the negative entropy term refines weight distribution to reduce reliance on noisy samples, while the denoising loss corrects gradients to remove noise, restoring data integrity. Together, these mechanisms ensure robust optimization in noisy environments.

## 4 Experiments

We carry out numerous experiments to evaluate the performance of DRGO, aiming to solve the following critical research questions:

- **RQ1**: How does the performance of DRGO compare to state-of-the-art methods in OOD recommendation?
- **RQ2**: How does DRGO mitigate the effects of random noise?
- **RQ3**: How do the components proposed in DRGO contribute to improving the model's efficiency in OOD generalization?
- **RQ4**: How do different hyperparameter settings affect the model's performance?

**Datasets**. We directly follow previous work [23] and conduct experiments under three common distribution shift scenarios (i.e., popularity shift, temporal shift, exposure shift.) to validate the performance of DRGO. The experiments are conducted on four commonly used datasets: Food[1], KuaiRec[2] Yelp2018[3], and Douban[4]. We provide detailed dataset information and processing details in the Appendix B.

**Baselines**. We compare DRGO with other SOTA methods: LigtGCN [4], SGL [31], SimGCL [37], LightGCL [1], InvPref [30], InvCF [40], DDRM [44], AdvInfoNCE [39], CDR [25] , AdvDrop [38], and DR-GNN [23]. Detailed information on these methods is given in the Appendix D. Detailed settings for the parameters of the DRGO model can be found in the Appendix C.

## 4.1 Overall Performance (R1)

In this section, we evaluate the effectiveness of our method through comparative experiments under three scenarios: distribution shifts caused by different factors and comparisons under IID (Independent

---

[1]https://www.aclweb.org/anthology/D19-1613/
[2]https://kuairec.com
[3]https://www.yelp.com/dataset
[4]https://www.kaggle.com/datasets/

**Table 1: Performance comparison of various methods on OOD Datasets.**

| | Food | | | | KuaiRec | | | | Yelp2018 | | | |
|---|---|---|---|---|---|---|---|---|---|---|---|---|
| | R@10 | N@10 | R@20 | N@20 | R@10 | N@10 | R@20 | N@20 | R@10 | N@10 | R@20 | N@20 |
| LightGCN (SIGIR2020) | 0.0234 | 0.0182 | 0.0404 | 0.0242 | 0.0742 | 0.5096 | 0.1120 | 0.4268 | 0.0014 | 0.0008 | 0.0035 | 0.0016 |
| SGL (SIGIR2021) | 0.0198 | 0.0159 | 0.0324 | 0.0201 | 0.0201 | 0.4923 | 0.1100 | 0.4181 | 0.0022 | 0.0013 | 0.0047 | 0.0020 |
| SimGCL (WWW2022) | 0.0233 | 0.0186 | 0.0414 | 0.0269 | 0.0763 | 0.5180 | 0.1196 | 0.4446 | 0.0049 | 0.0028 | 0.0106 | 0.0047 |
| LightGCL (ICLR2023) | 0.0108 | 0.0101 | 0.0181 | 0.0121 | 0.0630 | 0.4334 | 0.1134 | 0.4090 | 0.0022 | 0.0015 | 0.0054 | 0.0026 |
| InvPref (KDD2022) | 0.0029 | 0.0014 | 0.0294 | 0.0115 | 0.0231 | 0.2151 | 0.0478 | 0.2056 | 0.0049 | 0.0030 | 0.0108 | 0.0049 |
| InvCF (WWW2023) | 0.0030 | 0.0012 | 0.0033 | 0.0013 | 0.1023 | 0.2242 | 0.1034 | 0.2193 | 0.0016 | 0.0008 | 0.0013 | 0.0008 |
| AdvInfoNCE (NIPS2023) | 0.0227 | 0.0135 | 0.0268 | 0.0159 | 0.1044 | 0.4302 | 0.1254 | 0.4305 | 0.0047 | 0.0024 | 0.0083 | 0.0038 |
| CDR (TOIS2023) | 0.0260 | 0.0195 | 0.0412 | 0.0254 | 0.0570 | 0.2630 | 0.0860 | 0.2240 | 0.0011 | 0.0006 | 0.0016 | 0.0008 |
| DDRM (SIGIR2024) | 0.0039 | 0.0036 | 0.0077 | 0.0049 | 0.0011 | 0.0158 | 0.0023 | 0.0128 | 0.0005 | 0.0004 | 0.0016 | 0.0007 |
| AdvDrop (WWW2024) | 0.0240 | 0.0251 | 0.0371 | 0.0237 | 0.1014 | 0.3290 | 0.1214 | 0.3289 | 0.0027 | 0.0017 | 0.0049 | 0.0024 |
| DR-GNN (WWW2024) | 0.0246 | 0.0205 | 0.0436 | 0.0279 | 0.0808 | 0.5326 | 0.1266 | 0.4556 | 0.0044 | 0.0029 | 0.0076 | 0.0041 |
| **Ours** | **0.0293** | **0.0289** | **0.0497** | **0.0310** | **0.1149** | **0.6837** | **0.1970** | **0.6367** | **0.0086** | **0.0045** | **0.0168** | **0.0068** |
| Improv. | 12.69% | 15.14% | 13.99% | 11.11% | 10.06% | 28.37% | 55.60% | 39.75% | 75.51% | 50.00% | 55.56% | 38.77% |

**Table 2: Performance comparison of various methods on IID Datasets.**

| Methods | Food | | | | KuaiRec | | | | Yelp2018 | | | |
|---|---|---|---|---|---|---|---|---|---|---|---|---|
| | R@10 | N@10 | R@20 | N@20 | R@10 | N@10 | R@20 | N@20 | R@10 | N@10 | R@20 | N@20 |
| LightGCN (SIGIR2020) | 0.0317 | 0.0235 | 0.0517 | 0.0240 | 0.0154 | 0.0174 | 0.0272 | 0.0210 | 0.0527 | 0.0684 | 0.0910 | 0.0807 |
| SGL (SIGIR2021) | 0.0231 | 0.0187 | 0.0374 | 0.0236 | 0.0414 | 0.0508 | 0.0675 | 0.0568 | 0.0639 | 0.0838 | 0.1082 | 0.0974 |
| SimGCL (WWW2022) | 0.0237 | 0.0197 | 0.0373 | 0.0303 | 0.0446 | 0.0466 | 0.0697 | 0.0531 | 0.0663 | 0.0865 | 0.1122 | 0.1057 |
| LightGCL (ICLR2023) | 0.0103 | 0.0091 | 0.0166 | 0.0111 | 0.0200 | 0.0559 | 0.0620 | 0.0664 | 0.0579 | 0.0766 | 0.0972 | 0.0885 |
| InvPref (KDD2022) | 0.0281 | 0.0202 | 0.0454 | 0.0259 | 0.0050 | 0.0121 | 0.0082 | 0.0117 | 0.0395 | 0.0513 | 0.0665 | 0.0597 |
| InvCF (WWW2023) | 0.0425 | 0.0261 | 0.0410 | 0.0251 | 0.0393 | 0.0600 | 0.0383 | 0.0650 | 0.0958 | 0.0866 | 0.0941 | 0.0856 |
| AdvInfoNCE (NIPS2023) | 0.0514 | 0.0298 | 0.0518 | 0.0300 | 0.0250 | 0.0163 | 0.0248 | 0.0162 | 0.1068 | 0.0975 | 0.1075 | 0.0976 |
| CDR (TOIS2023) | 0.0278 | 0.0210 | 0.0449 | 0.0267 | 0.0387 | 0.0398 | 0.0576 | 0.0459 | 0.0406 | 0.0540 | 0.0673 | 0.0616 |
| DDRM (SIGIR2024) | 0.0233 | 0.0200 | 0.0498 | 0.0299 | 0.0200 | 0.0407 | 0.0342 | 0.0455 | 0.0544 | 0.0632 | 0.0876 | 0.0800 |
| AdvDrop (WWW2024) | 0.0431 | 0.0258 | 0.0469 | 0.0276 | 0.0195 | 0.0172 | 0.0223 | 0.0189 | 0.0593 | 0.0531 | 0.0586 | 0.0524 |
| DR-GNN (WWW2024) | 0.0307 | 0.0226 | 0.0505 | 0.0291 | 0.0101 | 0.0119 | 0.0187 | 0.0147 | 0.0610 | 0.0787 | 0.1020 | 0.0911 |
| **Ours** | **0.0530** | **0.0311** | **0.0524** | **0.0323** | **0.0471** | **0.0647** | **0.0719** | **0.0683** | **0.1123** | **0.1011** | **0.1205** | **0.1106** |
| Improv. | 3.11% | 4.36% | 1.15% | 7.67% | 5.61% | 7.83% | 3.17% | 2.86% | 5.15% | 3.69% | 7.39% | 4.64% |

and Identically Distributed) conditions. Best results are displayed in **bold**, while second-best results are underlined.

**Evaluations on Popularity Shift**: Table 1 shows baseline algorithms handling popularity shifts on the Yelp2018 dataset. DRGO significantly outperforms recent benchmark algorithms, achieving a maximum improvement of 75.71%. These results validate DRGO's effectiveness. Contrastive learning-based methods like SimGCL and AdvInfoNCE also outperform other models due to their ability to enhance relative differences between users and items, mitigating popularity bias and improving recommendation diversity. Invariant learning methods show performance gaps as they rely on reliable environment labels. Diffusion-based methods using LightGCN (e.g., DDRM) underperform compared to LightGCN, indicating their unsuitability for handling popularity shifts.

**Evaluations on Temporal Shift**: Temporal shifts reflect changes in user interests over time, simulated by dividing training and test sets based on interaction timestamps in the Food dataset. DRGO consistently outperforms competitors, with a 13.23% improvement over the second-place model, DR-GNN, which uses the DRO theory. This approach addresses uncertainty in data distributions, enabling robust performance across different environments beyond relying on training data alone.

**Evaluations on Exposure Shift**: On the KuaiRec dataset, where the test set is missing completely at random and the validation set is not, DRGO consistently outperforms competitors, demonstrating its effectiveness in addressing OOD generalization across graphs in inductive learning. Compared to the best baseline, DRGO achieves performance improvements of 10.06%, 27.37%, 55.60%, and 39.75% across four metrics, respectively. The DR-GNN method also surpasses other baselines, highlighting the efficiency of DRO methods in enhancing recommendation system generalization for OOD data.

**Evaluations on IID Data**: Table 2 presents comparative results on an IID dataset, where DRGO demonstrates an average improvement of 4.72% over the latest SOTA models. This performance edge is mainly due to the generative diffusion module, which smooths data distribution in the latent space by reducing noise progressively, and helps uncover latent correlations between user and item features. Additionally, the integration of user and item attributes further enhances performance. These results confirm our model's

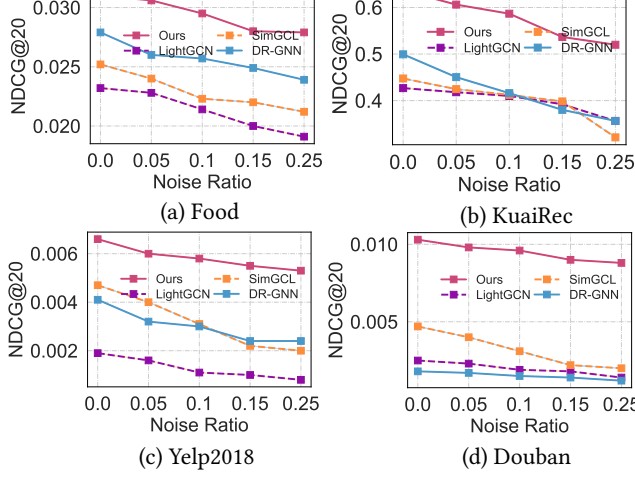

Figure 3: Relative performance decline with respect to noise ratio. We simulate different levels of noise by substituting 5%, 10%, 15%, and 25% of the interaction edges with artificial edges.

Table 3: Ablation Study on Food and Yelp2018.

| Method | Food | | Yelp2018 | |
|---|---|---|---|---|
| | Recall | NDCG | Recall | NDCG |
| LightGCN | 0.0404 | 0.0242 | 0.0035 | 0.0016 |
| SimGCL | 0.0414 | 0.0269 | 0.0106 | 0.0047 |
| LightGCL | 0.0181 | 0.0121 | 0.0054 | 0.0026 |
| DRGO w/o Diff. | 0.0448 | 0.0213 | 0.0140 | 0.0060 |
| DRGO w/o Reg. | 0.0476 | 0.0281 | 0.0141 | 0.0061 |
| DRGO w/o Feat. | 0.0476 | 0.0280 | 0.0151 | 0.0063 |
| LightGCL w/ DRGO | 0.0442 | 0.0253 | 0.0142 | 0.0058 |
| SimGCCL w/ DRGO | **0.0500** | **0.0327** | **0.0172** | **0.0070** |
| DRGO | 0.0497 | 0.0310 | 0.0168 | 0.0068 |

superior generalization to OOD data while validating its effectiveness on IID data.

## 4.2 Robustness Analysis (R2)

We investigated the robustness of DRGO and several baseline models against noisy data in recommendation systems. To assess the impact of noise on model performance, we randomly replaced different proportions of real edges with fake edges and retrained the models using the corrupted graphs as input. Specifically, we replaced 5%, 10%, 15%, and 25% of the edges in the original graph with fake edges. The results are shown in Figure 3. DRGO experiences less performance degradation than the baseline models, even when the noise ratio reaches 25%, with DRGO still maintaining a significant performance advantage. We attribute this result to two main factors: DRGO utilizes a diffusion model to denoise the user-item interaction graph, producing denoised embeddings that effectively reduce the proportion of noise concentrated in the uncertainty set. Second, we introduced a regularization term in the objective function to constrain outliers, thereby reducing the model's attention to these anomalous points. In the ablation study, we will analyze in

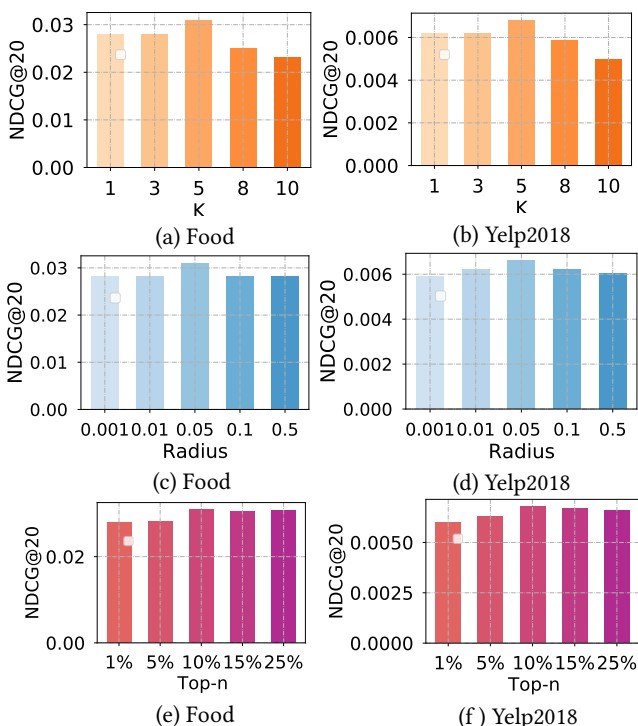

Figure 4: Analysis of the impact of various hyperparameters on model performance.

detail the contribution of these two components to the performance of GRGO. Additionally, we found that DR-GNN experienced a more significant performance drop compared to the other two baselines. This is mainly due to the inherent sparsity of these datasets. DR-GNN, while using DRO to improve OOD generalization, did not consider the impact of noise, making the noisy data have a more significant effect on the model's performance. Overall, the experimental results demonstrate that DRGO enhances the generalization performance in OOD recommendations and maintains robustness and effectiveness in noisy data.

## 4.3 Ablation Studies (R3)

We conduct experiments by individually removing three applied techniques and methods from DRGO: the diffusion denoising module (DRGO w/o Diff.), the regularization term for outlier constraints (DRGO w/o Reg.), and the primary user/item features (DRGO w/o Feat.) to validate the effectiveness of the proposed method. Meanwhile, we implement the DRGO method with LightGCL and SimGCL as backbones to verify the generality and portability of the proposed approach. These variants are retrained and tested on two datasets, and the results are shown in Table 5. From this, we draw the following significant conclusions: (1) After removing the diffusion module and the regularization term, the model's performance drops significantly, highlighting the importance of constraining noisy data and outlier values when using DRO to improve OOD generalization. Our proposed DRGO can prevent DRO from overfocusing on noisy data and outlier values during worst-case parameter learning, further enhancing recommendation performance on OOD data. Additionally, although the performance of DRGO

w/o Feat. also decreases, the drop is more minor than removing the other two modules, indicating that while user-item features contribute to performance improvement, they are not the primary factor driving the performance boost. (2) We implement DRGO with LightGCL and SimGCL as backbones, showing performance improvements. SimGCL w/ DRGO outperforms DRGO with Light-GCN as the backbone across all four metrics. We attribute this to SimGCL's unique method of enhancing embeddings, especially its approach of applying enhancement after denoising, which further boosts its performance. In conclusion, the ablation study highlights that each module in DR-GNN enhances the model's learning ability and validates the effectiveness and portability of DRGO.

### 4.4 Hyperparameter Analysis (R4)

**Impact of the number of clusters** $K$. We examine how varying the number of clusters, $K$, affects model performance in the context of constructing uncertainty sets. Results are presented in Figure 4 (a) and (b). On both datasets, model performance fluctuates as $K$ changes. Initially, as $K$ increases from 1 to 5, DRGO's performance improves, reaching an optimal level. However, further increases in $K$ lead to performance degradation. This may be because a smaller $K$ results in suboptimal worst-case selections, hindering generalization, while a larger $K$ introduces redundancy and overfitting.

**Impact of the robust radius** $\rho$. We explore how the robust radius $\rho$ influences model performance. Figures 4(c) and (d) depict the effects on the Food and Yelp2018 datasets as $\rho$ varies. It is evident that selecting an appropriate $\rho$ value is crucial for optimal performance, with the best results achieved at $\rho = 0.05$ for both datasets. Furthermore, DRGO shows greater sensitivity to $\rho$ on Yelp2018, likely due to more pronounced popularity shifts that complicate generalization.

**Impact of the nominal distribution top-n%**. We calculate the betweenness centrality for each node and select the Top-n% of nodes to form the nominal distribution, studying its impact on DRGO's performance. Figures 4 (e) and (f) show that increasing the selection proportion enhances DRGO's performance, peaking at the top 10% of nodes. Beyond this, performance stabilizes. A higher proportion can lead to longer computation times and potential overfitting. Thus, DRGO chooses the top 10% to strike a balance and form the ideal distribution.

## 5 Related Work

### 5.1 Graph-based Recommender Systems

Graph Neural Networks (GNNs) play a vital role in recommendation systems by effectively modeling high-order interaction signals between users and items using graph-based domain information [3, 9]. NGCF [28] uses Graph Convolutional Networks (GCNs) to learn high-order embeddings on bipartite graphs, while LightGCN [4] improves upon this by omitting feature transformations and non-linear activations, enhancing efficiency and recommendation performance. Approaches like KGAT [27] and CGAT [15] advance GNN-based models by incorporating knowledge graphs and attention mechanisms to better capture user preferences. Contrastive learning and data augmentation techniques have also gained traction, as seen in SGL [31], which employs three data augmentation strategies on interaction graphs to bolster recommendations

through contrastive learning. SimGCL [37], on the other hand, injects noise into embeddings to create positive and negative samples, improving accuracy without direct graph augmentation. Despite these successes, methods such as NGCF [28], LightGCN [4], and KGAT [27] often assume an IID (Independent and Identically Distributed) scenario for training and testing data, which limits their ability to generalize in OOD (Out-of-Distribution) contexts. Other strategies, like those employed in SGL [31] and SimGCL [37], target specific data shift types, which restricts their adaptability to more complex, real-world distributional challenges.

### 5.2 Diffusion for Recommendation

Recent advancements in diffusion methods have significantly broadened their application in recommendation systems, particularly in sample generation and representation learning. Notable works such as DiffRec [26] and Diff-POI [19] leverage diffusion for general and spatial recommendation tasks, respectively. The integration of diffusion models with graph neural networks (GNN) is explored in DiffKG [7] and DiffGT [36], which combine diffusion with data augmentation and transformer models to enhance knowledge graph learning and top-$k$ recommendations. RecDiff [10] and DiffNet++ [32] utilize diffusion to refine user-user graphs, improving social recommendation accuracy, while MCDRec [16] optimizes multimodal data processing. DiFashion [33] demonstrates diffusion's versatility by generating personalized user outfits. Despite these successes, diffusion-based approaches in recommendation systems still face challenges related to OOD data, which can limit their generalizability.

### 5.3 DRO based Recommendation

The application of Distributionally Robust Optimization (DRO) in recommendation systems has gained significant traction among researchers as they seek to address challenges of distribution shifts [11, 22, 23, 35, 43, 45]. The essence of DRO is to enhance model robustness by optimizing performance under the worst-case data distribution. For instance, PDRO [43] addresses popularity shifts, while DROS [35], RSR [45], and DRO [34] focus on improving the generalization of sequential recommendation models on OOD data. Moreover, DR-GNN [23] integrates DRO with graph-based methods to tackle distribution shifts in graph learning. Despite their effectiveness, the generalization performance of these models is easily affected by noisy samples in the data.

## 6 Conclusions

This paper employs theoretical analysis and experimentation to reveal the vulnerability of existing DRO-based recommendation methods to noisy data. Consequently, it proposes an innovative graph recommendation approach called DRGO, which incorporates a denoising diffusion process and an entropy regularization term to mitigate the impact of noisy data. Comparative experiments conducted across multiple datasets and settings demonstrate the model's effectiveness in addressing distribution shift issues and its robustness against noisy data.

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

# A PROOFS AND DERIVATIONS

## A.1 The proof of KL divergence

PROOF. Given the mathematical definition of KL divergence:

$$D_{KL}(P||Q) = \sum_x P(x) \log\left(\frac{P(x)}{Q(x)}\right), \tag{17}$$

where $P$ and $Q$ denote the different distributions, respectively. The training data distribution $P_{train}$ and the testing data distribution $P_{ts}$ do not overlap. For any test data point $x_{\text{test}} \sim P_{\text{test}}$, since $P_{\text{train}}(x_{\text{test}}) = 0$, we have:

$$\lim_{x \to 0} \log\left(\frac{P_{ts}(x)}{P_{train}(x)}\right) \to \infty, \tag{18}$$

we further analyze the following:

**Invisibility of the Test Distribution**: During training, the model has no access to data points that resemble the distribution of

the test set. Consequently, the optimized model cannot generalize to these unseen test data points.

**Inability to Handle OOD Data**: Since the training set lacks data points that match the distribution of the test data, the optimization process cannot leverage any information from these OOD data points, leading to severe degradation in the model's performance on the test data.

Therefore, KL divergence will make it difficult for the optimization objective to converge effectively. □

### A.2 Example

**Example**. Assume the training dataset consists of clean samples and noisy samples, where the clean sample set is $\{(u_i, i^+, i^-)\}_{i \in [N_c]}$ and the noisy sample set is $\{(u_j, j^+, j^-)\}_{j \in [N_o]}$. Here, $u$ represents the user, $i^+$ denotes a positive sample (an item liked by the user), $i^-$ denotes a negative sample (an item disliked by the user), and $N_c$ and $N_o$ are the numbers of clean samples and noisy samples, respectively. We assume that the labels for the noisy samples are more chaotic and may include incorrect positive and negative sample pairs. Given the standard BPR loss function as follows:

$$\mathcal{L}_{BPR} = - \sum_{u,i^+,i^-} \ln(\sigma(\hat{r}_{u,i^+} - \hat{r}_{u,i^-})), \quad (19)$$

Now, introduce sample weights, where the weight of clean samples is defined as $w_c(i)$, and the weight of noisy samples is defined as $w_o(j)$, satisfying:

$$\sum_{i=1}^{N_c} w_c(i) + \sum_{j=1}^{N_o} w_o(j) = 1 \quad (20)$$

The Weighted BPR Loss Function is Expressed as:

$$\mathcal{L}_{\text{Weighted BPR}} = - \sum_{i=1}^{N_c} w_c(i) \ln \left( \sigma(\hat{r}_{u,i^+} - \hat{r}_{u,i^-}) \right)$$
$$- \sum_{j=1}^{N_o} w_o(j) \ln \left( \sigma(\hat{r}_{u,j^+} - \hat{r}_{u,j^-}) \right), \quad (21)$$

where $\hat{r}_{u,i} = \theta^T x_{u,i}$ represents the predicted rating of item $i$ by user $u$, and $x_{u,i}$ is the feature vector.

Suppose $\hat{\theta}$ is the estimator obtained by minimizing the weighted BPR loss function, then the variance of the parameter $\theta$ can be expressed as:

$$\text{Var}[\hat{\theta}] = \mathbb{E}\left[ (\hat{\theta} - \mathbb{E}[\hat{\theta}])^2 \right]. \quad (22)$$

By differentiating the weighted BPR loss function with respect to $\theta$, we get:

$$\frac{\partial \mathcal{L}_{\text{Weighted BPR}}}{\partial \theta} = - \sum_{i=1}^{N_c} w_c(i) x_{u,i^+} \Delta\hat{r}_{u,i} \sigma(-\Delta\hat{r}_{u,i})$$
$$- \sum_{j=1}^{N_o} w_o(j) x_{u,j^+} \Delta\hat{r}_{u,j} \sigma(-\Delta\hat{r}_{u,j}), \quad (23)$$

where $\Delta\hat{r}_{u,i} = \hat{r}_{u,i^+} - \hat{r}_{u,i^-}$.

Assuming that sample weights are proportional to the loss, i.e., $w_o(j)$ is related to the variance $\sigma_o^2$ of noisy samples, and $w_c(i)$ is related to the variance $\sigma_c^2$ of clean samples, the variance of $\theta$ can

be approximated. Finally, the variance of the parameter $\theta$ can be Expressed as:

$$\text{Var}[\hat{\theta} \mid X_D]$$
$$= \frac{\sum_{i=1}^{N_c} (w_c(i))^2 (x_{u,i^+} \Delta\hat{r}_{u,i})^2 \sigma_c^2 + \sum_{j=1}^{N_o} (w_o(j))^2 (x_{u,j^+} \Delta\hat{r}_{u,j})^2 \sigma_o^2}{\left( \sum_{i=1}^{N_c} w_c(i) (x_{u,i^+} \Delta\hat{r}_{u,i})^2 + \sum_{j=1}^{N_o} w_o(j) (x_{u,j^+} \Delta\hat{r}_{u,j})^2 \right)^2}. \quad (24)$$

### A.3 The proof of Theorem 3.1

PROOF. Before formally beginning the proof, we introduce the following definitions :

**Definition 1**. (Lipschitz Continuity) A function $f : X \rightarrow \mathbb{R}^m$ is said to be G-Lipschitz continuous if for all $x, y \in X$, it holds that $\|f(x) - f(y)\| \leq G\|x - y\|$.

**Definition 2**. (Smoothness) A function $f : X \rightarrow \mathbb{R}$ is called L-smooth if it is differentiable on $X$ and its gradient $\nabla f$ is L-Lipschitz continuous, meaning $\|\nabla f(x) - \nabla f(y)\| \leq L\|x - y\|$ for all $x, y \in X$.

Next, we provide the proof that the BPR loss function $\mathcal{L}_{rec}$ in DRGO is G-Lipschitz continuous and L-smooth with respect to $x$. To facilitate the proof, we rewrite the BPR loss function. Assuming there is a user $u$ and items $i$ and $j$ (where $i$ is the positive sample and $j$ is the negative sample), the BPR loss function is typically defined as:

$$\mathcal{L}(f(u, i, j)) = - \log \left( \sigma(f(u, i) - f(u, j)) \right), \quad (25)$$

where $f(u, i)$ represents the predicted rating of user $u$ for item $i$, and generally $f(\cdot)$ is a dot product. $\sigma(x)$ is the sigmoid function.

**Lipschitz Continuity**. To prove that the BPR loss function is $G$-Lipschitz continuous, we need to show that there exists a constant $G > 0$, such that for all inputs $(u, i, j)$ and $(u', i', j')$:

$$|\mathcal{L}(f(u, i, j)) - \mathcal{L}(f(u', i', j'))| \leq G\|(u, i, j) - (u', i', j')\|, \quad (26)$$

consider the difference between the inputs $(u, i, j)$ and $(u', i', j')$:

$$|\mathcal{L}(f(u, i, j)) - \mathcal{L}(f(u', i', j'))|$$
$$= | - \log(\sigma(f(u, i) - f(u, j))) + \log(\sigma(f(u', i') - f(u', j')))|$$
$$\leq \left| \log(\sigma(f(u, i) - f(u, j))) - \log(\sigma(f(u', i') - f(u', j'))) \right|$$
$$\text{(the triangle inequality)}$$
$$\leq \frac{|(f(u, i) - f(u, j)) - (f(u', i') - f(u', j'))|}{\min(\sigma(f(u, i) - f(u, j)), \sigma(f(u', i') - f(u', j')))}, \quad (27)$$

The above step leverages the property of the sigmoid function $\sigma'(x) = \sigma(x)(1 - \sigma(x))$, which has a maximum value of 0.25 at any $x$, as well as the Lipschitz property of the logarithmic function $|\log a - \log b| \leq |a - b| / \min(a, b)$ to make the estimation. Assuming the rating function $f(u, i)$ is $K$-Lipschitz continuous with respect to the input $(u, i)$, then:

$$|f(u, i) - f(u', i')| \leq K\|(u, i) - (u', i')\|. \quad (28)$$

Thus, by choosing $G = 2K \times 0.25$, we establish the $G$-Lipschitz continuity of BPR.

**Smoothness**: To demonstrate that the BPR loss function is $L$-smooth, we need to show that its gradient is Lipschitz continuous. First, we calculate the gradient of the BPR loss function:

$$\frac{\partial \mathcal{L}(f(u, i, j))}{\partial f(u, i)} = - \frac{\partial}{\partial f(u, i)} \log(\sigma(f(u, i) - f(u, j))). \quad (29)$$

Using the chain rule and the property of the sigmoid function, we get:

$$\frac{\partial \mathcal{L}(f(u,i,j))}{\partial f(u,i)} = -\frac{1}{\sigma(f(u,i) - f(u,j))} \cdot \sigma'(f(u,i) - f(u,j)). \quad (30)$$

Since $\sigma'(x) = \sigma(x)(1 - \sigma(x))$, we have:

$$\frac{\partial \mathcal{L}(f(u,i,j))}{\partial f(u,i)} = -\frac{\sigma(f(u,i) - f(u,j))(1 - \sigma(f(u,i) - f(u,j)))}{\sigma(f(u,i) - f(u,j))}. \quad (31)$$

Simplifying, the gradient becomes:

$$\frac{\partial \mathcal{L}(f(u,i,j))}{\partial f(u,i)} = -(1 - \sigma(f(u,i) - f(u,j))). \quad (32)$$

Similarly, for $f(u,j)$, we compute:

$$\frac{\partial \mathcal{L}(f(u,i,j))}{\partial f(u,j)} = \sigma(f(u,i) - f(u,j)). \quad (33)$$

To prove that the gradient of the BPR loss function is Lipschitz continuous, we analyze the rate of change in the gradient. Consider two input pairs $(u,i,j)$ and $(u',i',j')$, and compute the difference in their gradients:

$$\|\nabla \mathcal{L}(f(u,i,j)) - \nabla \mathcal{L}(f(u',i',j'))\|. \quad (34)$$

Now, calculate the gradient difference for each component:

$$\left| \frac{\partial \mathcal{L}(f(u,i,j))}{\partial f(u,i)} - \frac{\partial \mathcal{L}(f(u',i',j'))}{\partial f(u',i')} \right|$$
$$= \left| (1 - \sigma(f(u,i) - f(u,j))) - (1 - \sigma(f(u',i') - f(u',j'))) \right| \quad (35)$$
$$= \left| \sigma(f(u',i') - f(u',j')) - \sigma(f(u,i) - f(u,j)) \right|.$$

Using the property of the sigmoid function, $\sigma(x)$ is 1-Lipschitz continuous (since $|\sigma'(x)| \le 0.25$ and $\sigma(x)$ is monotonically increasing):

$$\left| \sigma(f(u',i') - f(u',j')) - \sigma(f(u,i) - f(u,j)) \right|$$
$$\le \left| (f(u',i') - f(u',j')) - (f(u,i) - f(u,j)) \right|. \quad (36)$$

Assume that $f(u,i)$ and $f(u,j)$ are $K$-Lipschitz continuous with respect to their parameters:

$$\left| (f(u',i') - f(u',j')) - (f(u,i) - f(u,j)) \right|$$
$$\le K(\|u - u'\| + \|i - i'\| + \|j - j'\|). \quad (37)$$

Thus, we can conclude that the gradient is $K$-Lipschitz continuous. To further prove that the BPR loss $L$ is G-Lipschitz continuous and L-smooth, we can easily extend this to the optimization objective of DGRO, that is, $R(\theta, w_{q_i})$ is G-Lipschitz continuous and L-smooth. For any two points $\theta_1$ and $\theta_2$, the change in the function value is constrained by the following inequality:

$$|R(\theta_1, w_{qi}) - R(\theta_2, w_{qi})| \le G\|\theta_1 - \theta_2\|. \quad (38)$$

This implies that within a compact domain, the function does not vary too rapidly, thus helping to control the range of $R(\theta, w_{qi})$. The L-smooth property implies that the gradient of the function is Lipschitz continuous with constant $L$, indicating that the function exhibits quadratic growth and does not increase abruptly. Formally, this property is expressed as:

$$R(\theta_2, w_{qi}) \le R(\theta_1, w_{qi}) + \nabla R(\theta_1, w_{qi})^T (\theta_2 - \theta_1) + \frac{L}{2}\|\theta_2 - \theta_1\|^2. \quad (39)$$

This quadratic growth further constrains the rate at which the function can increase. **We further provide proof that DRGO is bounded by a constant B**. By combining the G-Lipschitz continuity and L-smoothness properties, we analyze each term in the DRGO loss. **For the first term**, $\mathcal{L}_{rec}$ is the BPR loss. For convenience in proving, we rewrite the BPR loss: The specific BPR loss can be expressed as:

$$\mathcal{L}_{rec}(f(x), y) = -\log(\sigma(f(x_u) - f(x_i))), \quad (40)$$

where $f(x_u)$ denotes the predicted score of user $u$ for the positive sample $x_u$, while $f(x_i)$ denotes the predicted score of user $u$ for the negative sample $x_i$. Since $\sigma(z)$ is the sigmoid function, with an output range of $(0, 1)$, the loss function $L_{rec}$ is inherently bounded. The upper bound of the BPR loss can be derived based on the properties of the sigmoid function:

$$-\log(\sigma(f(x_u) - f(x_i))) \le -\log(\sigma(0)) = -\log(0.5) = \log 2. \quad (41)$$

Therefore, for each $i$, it follows that $L_{rec}(f(x), y) \le \log 2$. The upper bound of the weighted sum is then given by:

$$\sum_{i=1}^{n} w_{qi} L_{rec}(f(x), y) \le \log 2 \sum_{i=1}^{n} w_{qi} = \log 2. \quad (42)$$

This holds because the sum of the weights $w_{qi}$ is equal to 1. **For the second term**, the form of the entropy regularization term is:

$$-\beta \sum_{i=1}^{n} w_{qi} \log w_{qi}. \quad (43)$$

Since $w_{qi}$ is a probability distribution, we have:

$$\sum_{i=1}^{n} w_{qi} = 1. \quad (44)$$

The entropy $H(w) = -\sum_{i=1}^{n} w_{qi} \log w_{qi}$ reaches its maximum when all $w_{qi}$ are equal, i.e., $w_{qi} = \frac{1}{n}$. At this point, the maximum value of the entropy is:

$$H(w) \le \log n \quad (45)$$

Therefore, the upper bound of the regularization term is:

$$-\beta \sum_{i=1}^{n} w_{qi} \log w_{qi} \le \beta \log n. \quad (46)$$

**For the third term**, the denoising loss $L_{denoising}(G_0)$ depends on the nature of the denoising generative model $G_0$. Assume that $G_0$ is a smooth and bounded generative model, and its domain is also bounded. In this case, the denoising loss function will also be constrained within a certain constant range. Let this constant be $C_{denoise}$, then:

$$L_{denoising}(G_0) \le C_{denoise}. \quad (47)$$

Thus, we can conclude that the overall upper bound of the function is:

$$R(\theta, w_{qi}) \le \log 2 + \beta \log n + C_{denoise}. \quad (48)$$

Let this expression be denoted as a constant $B$, i.e.,

$$R(\theta, w_{qi}) \le B. \quad (49)$$

Thus, we have proven that the function $R(\theta, w_{qi})$ is bounded by the constant $B$. We analyze the generalization error using Rademacher complexity, mainly focusing on the complexity of the BPR loss in the hypothesis class. For a sample set $S = \{(x_i, y_i)\}_{i=1}^{n}$ and a

**Table 4: Detailed statistics for each dataset.**

| Dataset | #Users | #Items | #Interactions | Density |
|---------|--------|--------|---------------|---------|
| Food | 7,809 | 6,309 | 216,407 | $4.4 \times 10^{-3}$ |
| KuaiRec | 7,175 | 10,611 | 1,153,797 | $1.5 \times 10^{-3}$ |
| Yelp2018 | 8,090 | 13,878 | 398,216 | $3.5 \times 10^{-3}$ |
| Douban | 8,735 | 13,143 | 354,933 | $3.1 \times 10^{-3}$ |

hypothesis class $F$, the Rademacher complexity $\hat{R}^n(F)$ is defined as:

$$\hat{R}_n(F) = \mathbb{E}_\sigma \left[ \sup_{f \in F} \frac{1}{n} \sum_{i=1}^{n} \sigma_i f(x_i) \right], \tag{50}$$

where $\sigma_i \in \{-1, 1\}$ are Rademacher random variables used to measure the model's ability to fit random noise. Assuming the Rademacher complexity is $\hat{R}^n(F)$, the generalization error bound based on the Rademacher complexity for the BPR loss function is:

$$\mathbb{E}_Q \left[ \frac{1}{n} \sum_{i=1}^{n} w_{qi} L_{rec}(f(x_i), y_i) \right]$$
$$\leq \frac{1}{n} \sum_{i=1}^{n} w_{qi} L_{rec}(f(x_i), y_i) + 2\hat{R}_n(F) + \sqrt{\frac{ln(1/\sigma)}{2n}}, \tag{51}$$

where $\sigma \in (0, 1)$ This indicates that the gap between the error on the training set and the generalization error is controlled by $\hat{R}_n(F)$. Furthermore, we have:

$$\hat{R}(\theta, w_{q_i}) \leq R(\theta, w_{q_i}) + 2\hat{R}_n(F) + BW(P_{train}, Q) + B\sqrt{\frac{ln(1/\sigma)}{2n}}. \tag{52}$$

The upper bound of the generalization risk on $Q$ is mainly determined by its distance to $P_{train}$: $W(P_{train}, Q)$. □

## B Dataset Information and Processing Details

Table 4 provides the statistics of each dataset, detailing the number of users, items, interactions, and the dataset's sparsity. A brief introduction to all datasets is as follows:

- **Food**. This dataset includes over 230,000 recipes and millions of user interactions, such as reviews and ratings, making it useful for analyzing user preferences and building food-related recommendation systems.
- **KuaiRec**. The KuaiRec dataset is a fully-observed dataset from the Kuaishou app, containing millions of dense user-item interactions with minimal missing data. It is ideal for studying recommendation systems, including the effects of data sparsity and exposure bias.
- **Yelp2018**. This dataset contains user reviews, ratings, and business information, commonly used for recommendation systems and user behavior analysis.
- **Douban**. A Chinese social media platform, including user ratings, reviews, and social network connections. It is often used for research on recommendation systems, social influence, and collaborative filtering.

**Processing Details**. We retain only those users with at least 15 interactions on the Food dataset, at least 25 interactions on the Yelp2018 and Douban datasets, and items with at least 50 interactions on these datasets. For all three datasets, only interactions

with ratings of 4 or higher are considered positive samples. For the KuaiRec dataset, interactions with a watch ratio of 2 or higher are considered positive samples.

We will process the above dataset to construct three common types of OOD.

- **Popularity shift**: We randomly select 20% of interactions to form the OOD test set, ensuring a uniform distribution of item popularity. The remaining data is split into training, validation, and IID test sets in a ratio of 7:1:2, respectively. This type of distribution shift is applied to the Yelp2018 and Douban datasets.
- **Temporal shift**: We sort the dataset by timestamp in descending order and designate the most recent 20% of each user's interactions as the OOD test set. The remaining data is split into training, validation, and IID test sets in a ratio of 7:1:2, respectively. The food dataset is used for this type of distribution shift.
- **Exposure shift**: In KuaiRec, the smaller matrix, which is fully exposed, serves as the OOD test set. The larger matrix collected from the online platform is split into training, validation, and IID test sets in a ratio of 7:1:2, respectively, creating a distribution shift.

## C Hyper-parameters Settings

We implement our CausalDiffRec in Pytorch. All experiments are conducted on a single RTX-4090. Following the default settings of the baselines, we expand their hyperparameter search space and tune the hyperparameters as follows:

- Weight decay $\beta$: [0.1, 0.001, 0.0001, 0.00001].
- Number of GNN layers $L$: [1, 2, 3, 4, 5].
- Embedding size $d$: [16, 32, 64, 128].
- Number of cluster $K$: [1, 3, 5, 8, 10].
- Roubust radius $\rho$: [0.001, 0.01, 0.05, 0.1, 0.5].
- Top-n% Betweenness Centrality Ranking: [1%, 5%, 10%, 15%, 25%].
- Maximum diffusion steps $T$: [20, 50, 100, 200, 500].

For the parameters of the baseline models, we search for their optimal parameters based on the range of parameter options provided in their respective papers.

## D Baselines

We evaluate CausalDiffRec against these leading models:

- **LightGCN** [4]: An efficient collaborative filtering model utilizing graph convolutional networks (GCNs), streamlining NGCF's message propagation by removing non-linear projection and activation.
- **SGL** [31]: Builds upon LightGCN and incorporates structural augmentations to improve representation learning.
- **SimGCL** [37]: Implements a straightforward contrastive learning (CL) strategy, avoiding graph augmentations by introducing uniform noise in the embedding space for contrastive views.
- **LightGCL** [1]: Utilizes LightGCN as the foundation, adding uniform noise to the embedding space for contrastive learning without using graph augmentations.
- **CDR** [25]: Employs a temporal variational autoencoder to capture preference shifts and learns sparse influences from various environments.

**Algorithm 1** Training Procedure of DRGO

1: **Input:** The user-item interaction graph $\mathcal{G}(\mathcal{V}, \mathcal{E})$ and node feature matrix $\mathcal{X}$; The number of cluster $K$, the number of layers $L$, and the learning rate $\eta$.
2: **while** not converged **do**
3:   **for** all $k \in \{1, 2, \ldots, K\}$ **do**
4:     Get the denoising embeddings by Eq. (6) and Eq. (4);
5:     Calculate the nominal distribution $Q$ by Eq. (9) and the uncertainty sets $P_{train}$ is calculated by Eq. (10)
6:     // Forward process
7:     Calculate the weights of different groups $w_{qi}$ in Eq. (12)
8:     // Reverse process
9:     Calculate the gradients w.r.t. the loss in Eq. (12);
10:   **end for**
11:   Average the gradients over $|U|$ users and $K$ environments;
12:   Update the model $\theta$ via AdamW optimizer;
13: **end while**
14: **Output:** : Trained model parameter $\theta^*$.

- **InvPref** [30]: A general debiasing framework that iteratively separates invariant and variant preferences from biased user behaviors by estimating different latent bias environments.
- **InvCF** [40]: Aims to reduce popularity shift to discover disentangled representations that accurately reflect latent preferences and popularity semantics without assumptions about popularity distribution.
- **AdvInfoNCE** [39]: A variant of InfoNCE improving the recommender's generalization via a detailed hardness-aware ranking criterion.
- **DDRM** [44]: It is a model-agnostic denoising diffusion recommendation model designed to enhance the robustness of user and item representations from any recommender system, effectively mitigating the impact of noisy feedback.
- **AdvDrop** [38]: Addresses general and amplified biases in graph-based collaborative filtering through embedding-level invariance from bias-related views.
- **DR-GNN** [23]: A GNN-based OOD recommendation approach addressing data distribution shifts with Distributionally Robust Optimization theory.

# E Introduction to Diffusion Models

Diffusion models [6], due to their high-quality generation, training stability, and solid theoretical foundation, have achieved notable advancements in computer vision. These models are a type of deep generative model that operates through two distinct phases: the forward process and the reverse process.

**Forward process**. Given a real data distribution $p(x_0)$, the goal of the forward phase is to progressively add Gaussian noise of varying scales to the data $x_0$, ultimately obtaining a data point $x_t$ after $T$ steps of noise addition. In detail, adding noise from $x_{t-1}$ to $x_t$ is shown as:

$$q(x_t|x_{t-1}) = \mathcal{N}\left(x_t; \sqrt{1-\beta_t}x_{t-1}, \beta_t \mathbf{I}\right), \tag{53}$$

where $\beta_t \in (0, 1)$ controls the level of the added noise at step $t$. $\mathbf{I}$ denotes the identity matrix, and $\mathcal{N}$ represents the is the Gaussian

**Table 5: Performance comparison on the Douban dataset**

| Method | OOD | | IID | |
|---|---|---|---|---|
| | Recall | NDCG | Recall | NDCG |
| LightGCN | 0.0049 | 0.0019 | 0.0491 | 0.0503 |
| SGL | 0.0047 | 0.0020 | 0.0559 | 0.0574 |
| SimGCL | 0.0167 | 0.0073 | 0.0588 | 0.0612 |
| LightGCL | 0.0113 | 0.0050 | 0.0495 | 0.0522 |
| InvPref | 0.0093 | 0.0038 | 0.0144 | 0.0161 |
| InvCF | 0.0033 | 0.0013 | 0.0579 | 0.0606 |
| AdvInfoNCE | 0.0103 | 0.0053 | 0.0622 | 0.0647 |
| CDR | 0.0019 | 0.0009 | 0.0166 | 0.0180 |
| DDRM | 0.0012 | 0.0010 | 0.0522 | 0.0518 |
| AdvDrop | 0.0046 | 0.0021 | 0.0204 | 0.0213 |
| DR-GNN | 0.0038 | 0.0017 | 0.0538 | 0.0550 |
| DRGO | 0.0269 | 0.0103 | 0.0598 | 0.0634 |
| Impro. | 61.08% | 41.10% | 1.70% | 3.59% |

distribution which means $x_t$ is sampled from this distribution. By applying the reparameterization trick, $x_t$ can be directly derived from $x_0$, as demonstrated below:

$$q(x_t|x_0) = \mathcal{N}\left(x_t; \sqrt{\bar{\alpha}}x_0, (1 - \bar{\alpha})\mathbf{I}\right) \tag{54}$$

where $\bar{\alpha}_t = \prod_{i=1}^{t} \alpha_i$, $\alpha_i = 1 - \beta_i$.

**Reverse Process**. It aims to reconstruct the original data by training a model $p_\theta$ to approximate the reverse diffusion from $x_T$ to $x_0$. This process is governed by $p_\theta(x_{t-1} \mid x_t)$, where the mean $\mu_\theta$ and covariance $\Sigma_\theta$ are learned through neural networks. Specifically, the reverse process is defined as:

$$p_\theta(x_{t-1} \mid x_t) = \mathcal{N}\left(x_{t-1}; \mu_\theta(x_t, t), \Sigma_\theta(x_t, t)\right), \tag{55}$$

where $\theta$ represents the neural network's parameters. $\mu_\theta(x_t, t)$ and $\Sigma_\theta(x_t, t)$ denote the mean and covariance, respectively.

**Training**. The reverse process is optimized to minimize the variational lower bound (VLB), balancing reconstruction accuracy with model simplicity [29].

$$\mathcal{L}_{VLB} = \mathbb{E}_{q(x_{1:T}|x_0)} \left[ \sum_{t=1}^{T} D_{KL}(q(x_{t-1}|x_t, x_0) || p_\theta(x_{t-1}|x_t)) \right] \\ - \log p_\theta(x_0|x_1), \tag{56}$$

where $D_{KL}()$ denotes the KL divergence. As described in [6], to address the training instability in the model, we expand and reweight each KL divergence term in the VLB using a specific parameterization. Consequently, the mean squared error loss is given by:

$$\mathcal{L}_{\text{sample}} = \mathbb{E}_{t, \mathbf{x}_0, \epsilon_t} \left\| \epsilon_t - \epsilon_\theta\left(\sqrt{\bar{\alpha}_t}\mathbf{x}_0 + \sqrt{1 - \bar{\alpha}_t}\epsilon, t\right) \right\|^2. \tag{57}$$

