# OpenReview forum: "Distributionally Robust Graph Out-of-Distribution Recommendation via Diffusion Model"
_ACM.org/TheWebConf/2025/Conference — WWW 2025 Poster_

### Official Review · Reviewer_VReo · 2024-11-04

**Novelty:** 4
**Technical Quality:** 5

**Review:**

This paper presents a novel approach for Graph Neural Network-based recommendation systems in out-of-distribution (OOD) scenarios, called the Distributionally Robust Graph model for OOD Recommendation (DRGO). The key innovation lies in employing a diffusion model to denoise embeddings in the latent space, along with an entropy regularization term during optimization to mitigate the impact of noise. The authors validate DRGO's effectiveness through comparative experiments on four real-world datasets.

Positive aspects

1. The attempt to combine Distributionally Robust Optimization (DRO) with diffusion models to address noisy data into graph contrastive recommendation scenarios is a valuable attempt.

2. The authors provide theoretical proofs regarding generalization error and noise reduction, enhancing the credibility of the proposed DRGO method.

3. The paper includes comprehensive experiments across multiple datasets and under different distribution shifts (e.g., popularity, temporal, and exposure shifts) that clearly demonstrate the model's effectiveness and robustness.

Negative aspects

1. The description of Sinkhorn DRO in this paper is limited to Section 3.1, but I find it difficult to understand how Sinkhorn functions within the model. I suggest providing a more detailed explanation of its application mechanism in the model.

2. In my opinion, this work proposes the use of diffusion models for denoising potential embeddings but combines it with a variational graph autoencoder (VGAE) for implementation, which may lead to some issues. Although VGAE and diffusion models share similarities in denoising capabilities, they differ in their theoretical foundations and practical applications. This could cause confusion about the modeling mechanisms. I suggest clarifying the respective roles of VGAE and diffusion modeling in this task to help readers better understand the principles behind the design.

3. Since this study uses diffusion models for recommendations, comparing it with the classic $ \textrm{DiffRec}^{[1]}$ would better showcase the proposed method’s improvements and advantages.

4. The sampling strategies for OOD and IID are challenging to understand. I suggest providing a more detailed explanation to help readers better grasp the design of the method.

5. The robustness and sensitivity of hyperparameters like the number of clusters, radius 𝑑, and centrality threshold require more detailed discussion, as the ablation and sensitivity analyses suggest potential variability in performance due to these settings.

References

[1] Wenjie Wang, Yiyan Xu, Fuli Feng, Xinyu Lin, Xiangnan He, and Tat-Seng Chua. 2023. Diffusion Recommender Model. In SIGIR. ACM, 832–841.

**Questions:**

1. What motivates the choice of Sinkhorn DRO over other divergence metrics in this specific application, and how does this impact computational efficiency compared to more conventional approaches?

2. Can the approach be adapted to include side information, such as temporal or contextual data, for more dynamic recommendations?

3. Why does the Yelp2018 dataset used in this paper differ significantly in user, item, and interaction counts (8090, 13878, 398216) from previous studies (31668, 38048, 1561406)?

4. Although the existing work is not closely related to graph contrastive learning-based approaches, I am puzzled by the performance of SimGCL in Table 2. I recommend that the authors re-evaluate this model's performance on these three datasets to ensure the accuracy of the experimental results.

**Reviewer Confidence:**

3: The reviewer is confident but not certain that the evaluation is correct

**Scope:**

4: The work is relevant to the Web and to the track, and is of broad interest to the community

---

### Official Review · Reviewer_Tnbe · 2024-11-28

**Novelty:** 3
**Technical Quality:** 3

**Review:**

**Summary:**

The paper proposes a method, DRGO, a distributed robust graph model for OOD recommendation. The proposed method first adopts a diffusion model to reduce the impact of noise in the latent space. In addition, an entropy regularization term is added for robust optimization. Sinkhorn DRO is used instead of KL-based DRO. Experimental results on several datasets show that DRGO outperforms existing methods.


**Strengths:**
1. Overall, the paper is well-written with a clear presentation.
2. Investigating OOD recommendations is a significant problem and direction.
3. The theoretical analysis is comprehensive.
4. Extensive experiments have been conducted to evaluate the performance of the proposed framework.

**Weakness:**
1. The proposed method optimized the VGAE and diffusion model together, which seems confusing. The diffusion model is for multiple-step generation, while VGAE is for single-step generation. The former is an improved variant of the latter, but when optimized together, the optimization goal of both is actually to restore the original embedding, which makes the overall optimization goal very confusing and reduces the credibility of the results.

2. In the pipeline, the process of VGAE goes from graph to embedding and then to graph. The input of diffusion is the midway embedding, which depends on VGAE. If firstly add noise to this embedding, then if the VGAE itself is not well trained, the generated embedding will be of poor quality. It is not reasonable to learn denoising on such a low-quality embedding.

3. There is a lack of experiments related to the hyperparameters of the diffusion model (such as the number of steps to add noise during training and the number of steps during inference).

4. In the method, the embedding obtained by the diffusion model is used as the initialization for the embedding of the last graph convolutional model (LightGCN), and then the recommendation loss is used to optimize the embedding. So, the authors should do some comparative experiments with random initialization or some common embedding initialization methods to prove that the adopted VGAE and diffusion model are effective, rather than relying on the contribution of recommendation loss of LightGCN.

5. This method adopts user/item features to add to the GCN, while many baselines used for comparison, such as LightGCN, LightGCL, and SimGCL, generally do not consider these features as input, but only use the interaction graph between users and items. If the baselines in this paper’s experiments do not add these features, the comparison will be unfair.

6. The legends in Figure 3 are not well organized, they can be placed outside the figure, like Figure 1.

7. There are some unexpected white blocks on the bars in Figure 4.

**Questions:**

Please refer to weakness.

**Reviewer Confidence:**

3: The reviewer is confident but not certain that the evaluation is correct

**Scope:**

3: The work is somewhat relevant to the Web and to the track, and is of narrow interest to a sub-community

---

### Official Review · Reviewer_5Lx7 · 2024-12-02

**Novelty:** 4
**Technical Quality:** 6

**Review:**

## Summary
The paper addresses limitations in existing Distributionally Robust Optimization (DRO)-based graph neural network (GNN) methods for recommendations, particularly their susceptibility to noisy data, which undermines out-of-distribution (OOD) generalization. The challenge lies in these models assigning excessive weight to noisy samples and relying on assumptions, such as overlapping distributions, that do not hold in realistic settings. The proposed DRGO employs a denoising diffusion model to mitigate noise in user-item interactions and introduces an entropy regularization term to balance sample weights. It further replaces the commonly used Kullback-Leibler divergence with Sinkhorn distance to handle non-overlapping distributions, enhancing robustness and generalization across both IID and OOD scenarios.

## Pros:
1. Effective use of a diffusion-based module to denoise interaction graphs, reducing the adverse effects of noisy training samples.
2. Replacing KL divergence with Sinkhorn distance allows optimization even in cases with minimal overlap between training and testing distributions.
3. The preliminary experiments for weight visualization and the Impact of noise are interesting and paper writing is well-motivated.
4. Comprehensive theoretical analysis with proven generalization bounds and strong experimental performance on multiple datasets under varying distribution shifts.

# Cons:
1. The integration of diffusion models and Sinkhorn DRO may increase computational overhead, particularly in large-scale datasets.
2. Performance depends on selecting optimal values for hyperparameters like the radius of the uncertainty set and the proportion of nodes in the nominal distribution.
3. The overall pipeline and methodology are similar to prior work DR-GNN [1]. The technical contribution is further limited.

[1] Wang B, Chen J, Li C, et al. Distributionally Robust Graph-based Recommendation System[C]//Proceedings of the ACM on Web Conference 2024. 2024: 3777-3788.

**Questions:**

1. The integration of diffusion models and Sinkhorn DRO may increase computational overhead, particularly in large-scale datasets. The data processing steps involve filtering out nodes with 15, 25, and 50 interactions. Can you detail the dataset size after filtering?
2. The overall pipeline and methodology are similar to prior work DR-GNN [1]. Can you highlight the difference between DR-GNN and DRGO?
3. The experiments include LightGCN [2], SGL [3], SimGCL [4], and LightGCL [5]. Can you explain why the following three data-augmented graph models demonstrate lower performance compared to LightGCN in OOD settings?

[1] Wang B, Chen J, Li C, et al. Distributionally Robust Graph-based Recommendation System[C]//Proceedings of the ACM on Web Conference 2024. 2024: 3777-3788.

[2] He X, Deng K, Wang X, et al. Lightgcn: Simplifying and powering graph convolution network for recommendation[C]//Proceedings of the 43rd International ACM SIGIR conference on research and development in Information Retrieval. 2020: 639-648.

[3] Wu J, Wang X, Feng F, et al. Self-supervised graph learning for recommendation[C]//Proceedings of the 44th international ACM SIGIR conference on research and development in information retrieval. 2021: 726-735.

[4] Yu J, Yin H, Xia X, et al. Are graph augmentations necessary? simple graph contrastive learning for recommendation[C]//Proceedings of the 45th international ACM SIGIR conference on research and development in information retrieval. 2022: 1294-1303.

[5] Cai X, Huang C, Xia L, et al. LightGCL: Simple yet effective graph contrastive learning for recommendation[J]. arXiv preprint arXiv:2302.08191, 2023.

**Reviewer Confidence:**

3: The reviewer is confident but not certain that the evaluation is correct

**Scope:**

3: The work is somewhat relevant to the Web and to the track, and is of narrow interest to a sub-community

---

### Official Review · Reviewer_wucE · 2024-12-03

**Novelty:** 5
**Technical Quality:** 4

**Review:**

## Summary
This article presents the Distributionally Robust Graph model for OOD recommendation (DRGO) framework, designed to enhance the robustness and generalization of graph-based recommendation systems against out-of-distribution (OOD) data. Existing Distributionally Robust Optimization (DRO)-based methods often fail due to overfitting noisy training samples and relying on infrequent assumptions like overlapping training and test distributions. To tackle these challenges, DRGO introduces three innovations: a denoising diffusion mechanism to reduce noise in the latent space, an entropy regularization term to balance sample weights, and Sinkhorn DRO to handle non-overlapping distributions effectively.
The paper provides theoretical and experimental validation of DRGO's effectiveness. It demonstrates superior performance on four datasets across three distribution shift types: popularity, temporal, and exposure. DRGO outperforms state-of-the-art methods under both IID and OOD conditions, maintaining robustness even with high noise levels. An ablation study highlights the importance of the diffusion mechanism and entropy regularization, while comparisons with models like LightGCN and SimGCL confirm DRGO’s superior adaptability to distribution shifts. This innovative framework offers a robust solution for recommendation tasks in dynamic, noisy environments.

## Strengths
The proposed work presents an innovative approach to tackling out-of-distribution (OOD) challenges in graph-based recommendation systems. The methodology is grounded in solid theoretical foundations, and the authors effectively motivate their study by first proving the critical issues they aim to address. Specifically:
- The theoretical analysis is well-constructed, particularly the discussion on the infinity magnitude of KL divergence when training and test sets are disjoint and the impact of noisy samples on BPR preference modeling.
- The technical analysis of gradient dynamics showcases the effectiveness of the proposed framework.
- Experimental evaluation is extensive, covering various datasets, conditions, and baselines, which validates the framework.
- The paper introduces a novel application of Sinkhorn DRO and a denoising diffusion mechanism for graph-based recommendations, contributing to the field.

## Limitations
While the paper demonstrates strong theoretical and empirical merits, its presentation and organization suffer from notable weaknesses:
- The narrative lacks a cohesive storytelling flow, with concepts introduced abruptly and without sufficient logical connections, making the paper feel disjointed. For example, section 3.4 introduces critical mechanisms without tying them back to the broader framework.
- The presentation of concepts can be improved, particularly in Section 2.1, where the environmental factors are first described as increasing the differences between the training and test sets, followed by a definition of these factors. Additionally, the description of the terms in Equation 1 could be clarified, as the frequent use of full stops gives the impression that the terms are disconnected from each other.
- The authors do not emphasize Sinkhorn DRO in the introduction as a key contribution, leaving its purpose unclear initially. This weakens the framing of the paper's originality.
- The model presentation in Section 3 is delayed and could benefit from an earlier introduction followed by detailed explanations of each component, improving the reader's understanding.
- Some of the most crucial components, such as Eq. 13, are relegated to the appendix (as Eq. 57), limiting accessibility to core content.
- Certain technical details that impact reproducibility are missing, such as the process for generating fake edges to simulate noise in Section 4.2 and the backbone used for denoising in the DDRM baselines. Additionally, in Section 3.4, the paragraphs feel disconnected. For example, in line 532, the "noise mitigation mechanism" is introduced without adequate context or connection to the previous content. Furthermore, in lines 397-398, two phrases are presented with colons, but the paragraph lacks clarity regarding the content and the authors' intended explanation.
- The related work section does not adequately position DRGO within the broader landscape of graph-based recommendation advancements, limiting its contextual impact.
- Conclusions are too brief and fail to provide an outlook on future work or broader implications.

**Questions:**

- Could you provide more details on how fake edges were created in Section 4.2 and the backbone used for DDRM baselines to ensure reproducibility?

**Reviewer Confidence:**

3: The reviewer is confident but not certain that the evaluation is correct

**Scope:**

3: The work is somewhat relevant to the Web and to the track, and is of narrow interest to a sub-community